# Complications and Frequency of Surgical Treatment with AO-Type Hook Plate in Shoulder Trauma: A Retrospective Study

**DOI:** 10.3390/jcm11041026

**Published:** 2022-02-16

**Authors:** Seung-Jin Lee, Tae-Won Eom, Yoon-Suk Hyun

**Affiliations:** Department of Orthopaedic Surgery, Kangdong Sacred Heart Hospital, Seoul 05355, Korea; lsjgoodlsj@gmail.com (S.-J.L.); pega1030@naver.com (T.-W.E.)

**Keywords:** distal clavicle fracture, acromioclavicular joint dislocation, hook plate fixation, painful shoulder stiffness

## Abstract

We investigated the complications and frequency of hook plate fixation in patients with shoulder trauma. We reviewed 216 cases of hook plate fixation use at our hospital between January 2010 and May 2020. Finally, we included 76 cases of acute distal clavicle fracture (DCF) and 84 cases of acute acromioclavicular joint dislocation (ACD). We investigated all complications after hook plate use, bony union in the DCF group, and reduction loss in the ACD group. We defined painful shoulder stiffness (PSS) as aggravating resting pain with stiff shoulder, and pain on shoulder elevation (PSE) as continued shoulder pain on elevation without PSS before plate removal. PSS was managed with intra-articular steroid injections or manipulation with or without arthroscopic capsular release (ACR). PSS occurred in 36 and 33 cases of DCF and ACD, respectively. PSE occurred in 17 of 76 fractures and 13 of 84 dislocations. However, no iatrogenic rotator cuff injury was verified by magnetic resonance imaging in patients with PSS or PSE. Subacromial erosion in patients with hook plate fixation should be considered a sequela and not a complication because it is unavoidable in surgery with an AO-type hook plate. The most common complication was PSS, followed by PSE.

## 1. Introduction

Injuries to the acromioclavicular joint are common, representing approximately 9% of all shoulder traumas, and distal clavicle fractures (DCFs) account for approximately 10–30% of all clavicle fractures [1,2]. Neer classification is widely used for DCFs, and type II fractures are unstable because this type is characterized by detachment of the coracoclavicular ligaments from the medial portion of the clavicle [3]. Rockwood classification [4] is also commonly used for acromioclavicular joint injuries, and most acromioclavicular joint dislocations (ACDs) are classified as types III, IV, and V. Most unstable ACDs and severely displaced DCFs usually require surgical treatment. Although numerous surgical methods have been introduced for cases requiring surgical treatment in these two types of acute shoulder trauma, the comparative advantage of each method is controversial [5,6,7,8,9]. Among the various surgical modalities reported, AO-type hook plate fixation has been an effective alternative plate fixation method for unstable DCF and severe ACD and has been used to promote the natural healing of ligaments [9,10,11,12,13]. The design of the AO-type hook plate is interesting because the hook passes below the acromion posterior to the acromioclavicular joint; therefore, it does not interfere with the joint. The plate is fixed to the superior clavicle. The superiorly displaced clavicle in DCF and ACD can be tightly fixed, and early mobilization is possible [14]. Fracture stability and acromioclavicular joint biomechanics are maintained, allowing early postoperative mobility [9,11,15,16,17]. However, numerous complications reported with AO-type hook plate fixation offset the high success rate and fast rehabilitation. The most frequently reported complications in previous studies are subacromial impingement and acromial bony erosion [5,18,19,20,21,22]. Most authors of previous studies on hook plate fixation have recommended plates removal as soon as bony union is achieved to prevent complications such as impingement and acromial erosion [5,17,18,23,24,25].

When explaining possible surgical complications to the patient and obtaining informed consent, it is important to know the frequency and type of each complication. To date, previous studies have reported the frequency of complications with a hook plate, which differs from that reported based on our clinical experience [5,26]. The authors indicated that the most frequent complication after surgery using a hook plate was painful shoulder stiffness (PSS) or secondary frozen shoulder. In contrast, Oh et al. [5] and Asadollahi et al. [26] reported acromial bony erosion and impingement pain as the most common complications in their systemic review of DCFs. This study aimed to retrospectively analyze the type of complications and frequency of hook plate use in shoulder trauma.

## 2. Materials and Methods

The Institutional Review Board (IRB) of Kangdong Sacred Heart Hospital approved this study (IRB no. 2021-08-003-001). This retrospective study included consecutive patients treated at a single institution between January 2010 and May 2020. The inclusion criteria were acute ACD or DCF with no other concomitant injuries observed on preoperative shoulder magnetic resonance imaging (MRI) before surgery. The exclusion criteria were concomitant lesions in the same shoulder on MRI, such as rotator cuff tears or labral injury; simultaneous arthroscopic shoulder surgery; fracture on the same shoulder; and short follow-up period (<3 months after plate removal). Among the 216 cases that underwent hook plate fixation, 57 were excluded. Thus, this study included 75 cases of DCF and 84 cases of ACD.

All surgeries were performed by a single surgeon (Hyun YS) using two types of AO hook plates—AO clavicular hook plate (Synthes, Raynham, MA, USA) and VariAx lateral hook plate (Stryker, Selzach, Switzerland). The only difference between the two metal plates is the difference in the angle of the hook part. The angle of hook is 90 degrees in the AO clavicular hook plate and 110 degrees in the VariAx lateral hook plate. According to the PDF file from the official website of Stryker, the hook of VariAx lateral hook plate is anatomically pre-contoured to fit the inferior aspect of the acromion. In several previous studies, the authors emphasized the complications can be minimized by performing an anatomic fit of the plate during the procedure and found it necessary to bend the hook [14,27,28]. Pre-contouring of 20 degrees in the VariAx lateral hook plate is thought to be influenced or reflect these previous research results [29,30,31]. In order to match the hook part of the metal plate with the lower edge of the acromion as much as possible, the special fluoroscopic view technique we developed was utilized for the best fit between the hook and the acromion, which was helpful in reducing the inevitable acromial erosion [27]. During the surgical procedure for DCF, at least one screw was inserted into the lateral fractured fragment to fix it. If it was thought that the screw insertion was insufficient, the metal plate and the lateral fractured fragment were wound together with wire and fixed.

After surgery, all patients received intravenous patient-controlled analgesia for 2 days and were administered opioid analgesic drugs with the same schedule. All patients underwent a similar rehabilitation program. Pendulum exercises were allowed as soon as patient comfort permitted. Patients were discharged with a home rehabilitation program, and active shoulder range of motion (ROM) was allowed as tolerated. All patients visited outpatient clinics once a month after discharge to measure the range of joint movement using hand-held goniometers. Moreover, pain intensity was measured using the visual analog scale (VAS), and other aspects of pain were assessed. Simple radiographs were used to observe the progress of bone union of the fractures and the occurrence of other possible complications.

We investigated all complications after application and removal of the hook plate. Bony union in DCF, reduction loss in ACD, and any other complications were investigated using simple shoulder and clavicle radiographs and clinical records. The evaluation of the severity of pain (VAS) and shoulder ROM before plate removal revealed that many patients complained of resting pain with an aggravating nature and loss of shoulder ROM before plate removal. We defined PSS as severe aggravating shoulder pain accompanied by stiffness that worsened, not improved, 2–3 months after hook plate insertion. According to the definition of painful stiffness, we categorized pain as aggravating, night-time, or resting pain. In addition, the limit of motion was defined as passive by <120° of forward elevation and <30° of external rotation at the side of the shoulder. In cases reporting continued shoulder pain on active elevation and at rest before plate removal, follow-up MRI examination was performed to rule out any rotator cuff lesion. First, an intra-articular injection of 40 mg of triamcinolone under fluoroscopic guidance was attempted 3–4 months after surgery for the management of PSS. For PSS that did not respond to intra-articular injection, we attempted additional procedures during plate removal surgery during hospitalization. Manipulation was the next solution for the PSS. Manipulation was performed during plate removal surgery, and it was performed after plate removal to prevent iatrogenic fracture using a hook plate. The procedure was performed under interscalene brachial plexus anesthesia (manipulation under anesthesia [MUA]) for pain control, and rehabilitation exercise was immediately initiated. If the patient’s ROM did not recover or was satisfied, additional arthroscopic capsular release (ACR) was performed. After MUA and ACR, patients were instructed to immediately initiate rehabilitation exercises for the maintenance of recovered shoulder ROM after surgery. All patients who received MUA or ACR visited us within 1 week after discharge to ensure they understood the rehabilitation exercises and performed them properly. After the early outpatient visit within the first week, the same outpatient visit schedule was followed for all other patients without PSS.

We also identified that many patients showed pain on active elevation, although their discomfort did not meet the diagnostic criteria of PSS as defined above. Therefore, pain on shoulder elevation (PSE) was defined as patients who complained of continued pain when the arm was actively raised >90° but had no PSS during application of the hook plate. We performed MRI to verify iatrogenic rotator cuff injury in patients with PSE and PSS.

Radiological assessment of reduction and reduction loss of ACD was performed preoperatively, postoperatively, and at the final follow-up using plain radiographs, which were compared to radiographs of the normal side. The images were analyzed and standardized to assess the coracoclavicular distance (CCD, height [in percent] to the contralateral shoulder between the upper border of the coracoid process and the inferior cortex of the clavicle). Loss of reduction (LOR) was defined as a ≥50% increase over the unaffected CCD. Subacromial erosion or osteolysis was defined as a rod-shaped radio-lucent lesion observed at the same point where the hook was seated. All radiographs were evaluated for osteolysis of the undersurface of the acromion, and erosion is well identified on axillary view of shoulder. It is difficult to accurately quantify subacromial erosion using radiographs. After plate removal, the erosion was checked on the axillary radiograph of the shoulder to determine whether it was present.

The plates were removed 4–5 months after surgery in the ACD group and after confirmation of bony union on computed tomography by an official radiological specialist in the DCF group. In cases of incomplete fracture union but painful shoulder stiffness, pain control was attempted with intra-articular steroid injection. If there was no improvement, after explaining the situation to the patient and obtaining consent, painful stiffness was resolved with MUA or ACR while the metal plate was removed. The progress of bone union was followed up on an outpatient basis. After removing the plate, postoperative follow-ups were performed at 2 weeks; 1, 2, 3, and 6 months; and 1 year.

Descriptive statistics, including means and standard deviations, were calculated to compare patients with DCF and ACD. The chi-square test and Fisher’s exact test were used to compare the incidence of PSS, PSE, and acromial fracture. Using Fisher’s exact test, we calculated the difference in the incidence of PSS according to fracture union in patients with DCF, incidence of PSS according to the occurrence of LOR in patients with ACD, and incidence of PSS according to the presence or absence of acromial fracture in patients with DCF and ACD. Data analysis was performed using Statistical Package for the Social Sciences software (version 21.0; SPSS Inc., Chicago, IL, USA). Statistical significance was set at *p* < 0.05.

## 3. Results

This study included 76 patients with DCF and 84 patients with ACD, with Rookwood types III, IV, and V occurring in 46, 2, and 36 patients, respectively. Among the 76 patients with DCF, the male-to-female ratio was 56:20, mean age was 48.5 years, duration before plate removal was 32.7 weeks, and mean follow-up time after plate removal was 6.8 months. Among the 84 patients with ACD, the male-to-female ratio was 76:8, mean age was 50.0 years, duration before plate removal was 20.4 weeks, and mean follow-up time after plate removal was 7.4 months (Table 1).

PSS was noted in 36 (47.4%) patients with DCF and 33 (39.3%) patients with ACD (Table 2). There was no difference in the incidence of PSS between patients with DCF and ACD (*p* = 0.303). Nine cases of PSS responded to intra-articular steroid injection, and 60 cases of PSS showed improvement with additional MUA (20 cases of DCF and 28 cases of ACD) or ACR (seven cases of DCF and five cases of ACD) during plate removal.

PSE was noted in 17 of 75 (22.4%) patients with DCF and 13 of 84 (15.5%) patients with ACD. There was no difference in the incidence of PSE between patients with DCF and ACD (*p* = 0.265). Before removing the hook plate, none of the patients with POM or PSS showed iatrogenic rotator cuff lesions on MRI. However, capsular thickening of the axillary pouch with high signal intensity as a typical finding of frozen shoulder was observed in patients with PSS (Figure 1). After removing the plate with or without surgical release for PSS, such as MUA or ACR, there was no recurrence during the follow-up period.

Varying degrees of subacromial erosion occurred in all patients, and acromial fracture occurred in one case in the DCF group and seven cases in the ACD group (Figure 2 and Figure 3). Five (6.7%) cases showed incomplete union and one (1.3%) case showed peri-implant stress fracture in the DCF group (Figure 4 and Figure 5). Four of the five cases of incomplete union showed fracture union without additional surgery during the follow-up period, and one case did not heal at the final follow-up, although the patient did not show any discomfort with non-union (Figure 6). The peri-implant fracture healed with conservative management before plate removal. LOR was observed in nine patients with ACD. Before removing the hook plate, LOR occurred in six patients with ACD. Among these six patients, five showed acromial fracture and one showed serious subacromial erosion (Figure 7). After removing the hook plate, LOR occurred in three patients during the follow-up period. However, none of these patients with LOR among patients with ACD showed tenderness on the acromioclavicular joint or pain during the follow-up period. Regarding acromial fracture, one (1.3%, 1 of 36) case was observed in the DCF group and seven (8.3%, 7 of 33) cases were noted in the ACD group (Figure 4). There was no difference in the incidence of acromial fracture between patients with DCF and ACD (*p* = 0.066). All eight patients had fracture union without additional surgical treatment during the follow-up period.

Among nine cases of ACD with LOR, three (33.3%) showed PSS. There was no difference in the incidence of PSS compared to that in 75 ACD cases without LOR (40%; *p* = 0.76). Among five patients with incomplete fracture union, one (20%) showed PSS. There was no difference in the incidence of PSS compared to that in 70 DCF cases with fracture union (50%; *p* = 0.362). Among eight patients with acromion fracture (one in the DCF group and seven in the ACD group), there was no statistically significant difference in the incidence of PSS between patients with and without acromial fractures (Table 3).

## 4. Discussion

In our study, when using two types of AO hook plates for the treatment of DCF and ACD, the most common complication was PSS, which was reported in 69 of 159 (43%) patients. Some patients with PSS improved with intra-articular steroid injection before plate removal, and all other patients improved with manipulation or ACR during plate removal. PSE was observed in 18.9% patients, but there was no iatrogenic rotator cuff injury, which was verified through MRI in patients with PSS or PSE. None of the patients with PSS and PSE showed positive findings in the impingement test. Acromial fracture, peri-implant fracture, and most incomplete union cases showed fracture union with conservative treatment. Patients with LOR and non-union in the ACD group did not show significant pain during the follow-up period.

The biggest advantage of hook plates in acute shoulder trauma (ACD and DCF) is their high success rate and rehabilitation [15,32]. The disadvantage is that patients with hook plates require a second surgery for implant removal because of uncomfortable points, which are caused by the hook plates [5,18,24,25,26].

Good et al. [11] showed a 95% union rate for hook plate fixation for DCFS, while our study demonstrated a union rate of 93.3%. Di Francesco et al. [33] showed 88% successful healing for ACD, while our study demonstrated a rate of 89.3%.

However, numerous complications reported with AO-type hook plate fixation offset the high success rate and fast rehabilitation. The most frequently reported complications reported in previous studies are subacromial impingement, impingement in motion, and subacromial bony erosion [5,13,18,19,20,21,22].

While explaining to the patients the complications that may occur after surgery preoperatively, it is important to provide information on not only the types of complications but also the frequency of each complication. Preoperatively, it is a principle to explain to patients all possible complications after surgery, but it is not easy for patients to remember all the information provided by the doctors. Therefore, it is good to provide information in the order of clinical importance, but it may also be good to provide information in the order of the frequency of occurrence of each complication. To date, few systemic reviews have demonstrated the frequency of complications with hook plates in DCFs [5,26]. Oh et al. [5] reported that the overall complication rate with hook plate fixation in 162 cases of DCF was 40.7%; the most common complication after hook plate fixation was impingement in motion (18.5%), followed by plate migration (9.3%), subacromial hole widening (4.3%), non-union (1.9%), and stiffness (1.2%). Asadollahi at el. [26] reported that the most common complications were subacromial osteolysis or erosion (27%), acromioclavicular arthrosis (22%), and peri-implant fracture (22%). In our study, PSS was the most common complication in patients with DCF (36 of 75 cases, 48%), followed by PSE (17 of 75 cases, 22.4%). Plate migration was not evaluated in our study because it might vary depending on the shooting angle during a simple radiographic examination. If plate migration in previous studies indicates the migration of the hook part with subacromial bony erosion, plate migration with varying degrees may have occurred in all our patients because varying degrees of subacromial erosions was observed in all our patients. We believe that additional computed tomography may be necessary for the precise quantification of the extent of plate migration and subacromial erosion in all patients; however, we did not evaluate both.

First, we need to clarify the definition of complications that may occur after surgery or procedures. Clavien et al. [34] proposed that complications and sequelae result from procedures, adding new problems to the underlying disease. However, complications are unexpected events that are not intrinsic to the procedure, whereas sequelae are inherent to the procedure. Asadollahi et al. [26] reported subacromial erosion in 27% of patients, Sim et al. [35] reported erosion in 62% of ACD cases, and Oh et al. [13] reported subacromial erosion in 66.7% of DCF cases and 38.5% of ACD cases. Kim et al. [36] reported that subacromial erosion occurred in all acute ACD cases according to computed tomography findings, and the mean erosion depth was approximately 50% of the acromial thickness. Kim et al. [37] also reported that subacromial erosion was observed in all patients. In our study, varying degrees of subacromial erosion were observed in all patients with DCF and ACD. Subacromial bony erosions may be a sign of migration between the plate and the acromion in patients. Unstable DCF and ACD are fixed with a lever on the hook, and some degree of hook migration may be unavoidable [14]. This unwanted bony erosion by hook migration can be minimized by bending the tip of the plate on the acromion side, adjusting to the patient’s anatomical fit [9,19]. However, some degree of bony erosion is inevitable because the tip of the plate pressurizes the acromion upward. Unless the subacromial bony erosion becomes an acromial fracture or leads to PSE, this inevitable finding itself should be categorized as a sequela and not a complication. Otherwise, we must explain to the patient that surgical treatment using a hook plate causes complications in all cases, which seems inappropriate.

Impingement pain or pain on shoulder motion after hook plating was reported in 30/44 (68%) patients in a study by Renger et al. [38], 9/10 (90%) patients in a study by Bhangal et al. [39], 6/31 (19.3%) patients in a study by Meda et al. [17], in all 3 patients in a study by Chandrasenan et al. [40], 9/28 (32.1%) patients in a study by Tiren et al. [19], and 35/64 (54.7%) patients in a study by Hyun et al. [27]. Subacromial impingement and impingement in motion in previous studies may include concerns regarding iatrogenic rotator cuff injuries [10]. Chandrasenan et al. [40] and ElMaraghy et al. [16] reported that rotator cuff injuries could occur with the use of a hook plate. In our study, MRI examinations were performed to determine the cause of pain in all patients with PSS and PSE, and none of the patients showed iatrogenic rotator cuff injuries. The incidence rate of impingement in motion reported by Oh et al. [5] (18.5%) is comparable to that of PSE (18.9%) in our study, although it is unclear whether impingement in motion and PSE are the same. As reported by the other abovementioned authors, the symptoms of pain and loss of motion in our study disappeared after plate removal. Regarding PSE in our study, there may be some debate regarding whether it is a surgical complication. Two facts should be considered before concluding on this issue. First, PSE disappeared after plate removal because PSE might be related to hook contact. Both PSE and impingement in motion can be considered transient symptoms and not complications. However, PSE and impingement in motion may not disappear without plate removal. Second, PSE does not always occur in all patients undergoing hook plate fixation, but subacromial bony erosion is always observed in all patients.

Few previous studies have described painful shoulder stiffness or secondary frozen shoulder after hook plate use. Tiren et al. [19] reported impingement and subacromial osteolysis complaints in 32% and 25% patients, respectively. Their complaints were mild, and none of the patients developed a frozen shoulder or required early plate removal. Oh et al. reported only 1.2% stiffness in their systemic review [5], but they reported a high rate of incidence of stiffness (65%) in their recent case series [13]. The “stiffness” described in this previous study [13] did not include the pain, unlike PSS in our study. In our study, almost half of the patients (43.3%, 69 of 159 patients) with hook plates experienced PSS. Although they did not describe any other procedures for the management of stiffness except plate removal, MUA with or without ACR during plate removal was applied to painful stiffness that did not improve on intra-articular steroid injection before plate removal in our study. There was no recurrence of symptoms in PSS cases during the follow-up period.

Di Francesco et al. [33] reported 5 of 42 (12%) cases of LOR after 1 year follow-up of acute ACD cases. Cases of LOR occurred after plate removal. The five patients in whom the ligaments did not heal showed fair results on the Constant–Murley scale, with local pain increasing during activities in which the arm was raised above the head. In our study, among the nine (10.7%) cases of LOR, six (7.1%) were observed before plate removal and three (3.6%) were observed after plate removal in the acute ACD group. When we analyzed whether the presence of LOR was associated with the incidence of PSS in ACD cases, no association was found (Table 3).

As a rare complication after hook plate fixation, incomplete union in DCF cases and acromial fracture in DCF and ACD cases were observed in our study. The incidence of incomplete union or non-union in the DCF group was 4% in a study by Tiren et al. [19], 5% in a study by Good et al. [11], and 0% in a study by Lee et al. [41]. In our study, incomplete fracture union occurred in 6.7% of patients, and no patient complained of pain or tenderness and underwent additional surgery for bone union during the follow-up period after metal plate removal. In terms of the risk factor for PSS, incomplete union in DCF cases was not related to the incidence of PSS (Table 3).

An acromial fracture is probably a result of subacromial erosion. Several studies have reported the occurrence of acromion fractures at a weakened location [42,43,44]. Eight patients sustained acromion fractures in our study. In terms of the risk factors for PSS, acromial fracture cases were also not related to the incidence of PSS (Table 3). Therefore, incomplete fracture union and acromial fracture do not appear to affect the occurrence of PSS.

It is important not only to determine the type and frequency of complications, but also to try to reduce them. Almost all complications when using hook plates are due to improper contact between the hook and the undersurface of acromion. Variables related to hook angle or contact between hook and acromion may be thought to influence the type and frequency of complications observed. Several studies support the necessity of the bending of hook for the optimal fit or contact between the hook and the undersurface of acromion [14,16,28]. As part of this effort, studies have also been reported on whether the angle of the hook part is 90 degrees or whether a larger angle is appropriate. As a result, it was reported that a hook angle of 105 degrees or 110 degrees is appropriate to reduce the complications related to the hook [29,30,31]. Yoon et al. reported that the hook made a pin-point contact with the undersurface of the acromion, and the force concentration phenomenon associated with the hook plate of existing designs results from cases of morphological mismatch, such as excessive inclination and improper occupation of the subacromial space [45]. From the point of view that the contact between the hook and the lower acromion is related to all complications, it can be inferred that a wider surface than the pin-point contact is better. When considering both the angle of the hook and the contact surface between the hook and the undersurface of acromion, we would like to emphasize the customized bending of hook for the optimal contact between the hook and the undersurface of acromion. The angle between the long axis of the clavicle and the underside of the acromion will vary from person to person, therefore customized bending of hook under the appropriate X-ray view is thought to be better than uniform bending such as 105 or 110 degrees. In order to widen the contact surface of the hook and the acromion, it would be good to have an X-ray view that can evaluate the contact between these two structures as accurately as possible. In order to match the hook part of the metal plate with the undersurface of the acromion as much as possible, the special fluoroscopic view technique we developed was utilized for the best fit between the hook and the acromion in all our patients, which was helpful in reducing the inevitable acromial erosion [27]. Therefore, looking at the osteolysis of the lower acromion observed in our patients, it can be observed that wear or friction occurred in a rather large area rather than due to pin-point contact (Figure 2). From the point of view that the angle of the hook and the size of the contact surface between the hook and the acromion can affect the frequency and type of complications of the hook metal plate, it can be estimated that the two variables mentioned above are unlikely to act as variables in this study because the most appropriate method was equally used for all patients.

This study has some limitations. First, our study has limitations inherent to those of similar retrospective non-randomized studies. Second, we did not analyze the relationships between all complications. Our definitions of two complications (PSS and PSE) and postoperative complication can be debatable. Despite these limitations, our study provided useful data for patients’ understanding of the postoperative complications before undergoing surgical treatment with an AO-type hook plate.

## 5. Conclusions

Subacromial bony erosion occurred in all patients undergoing AO-type hook plate fixation, but it should be considered a sequela and not a postoperative complication because it is unavoidable in surgical treatment with an AO-type hook plate. The most common complication was shoulder stiffness. MUA with or without ACR during plate removal could relieve this painful shoulder stiffness refractory to intra-articular steroid injection.

## Figures and Tables

**Figure 1 jcm-11-01026-f001:**
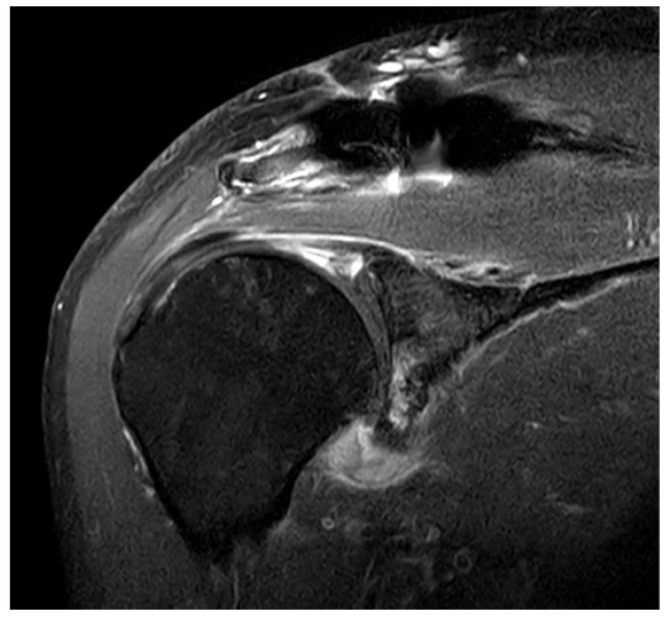
Typical findings on magnetic resonance imaging (MRI) in patients with painful shoulder stiffness. High signal intensity is observed in thickened joint capsule, which is emphasized on axillary capsular pouch.

**Figure 2 jcm-11-01026-f002:**
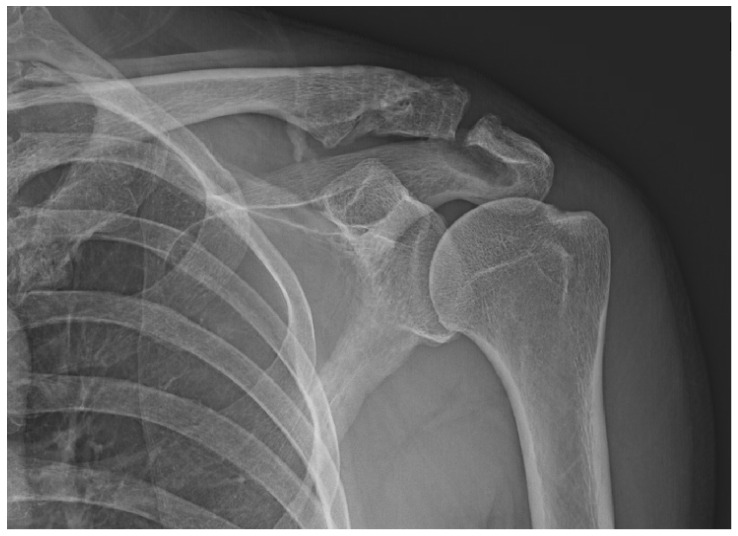
Acromial bony erosion. Varying degrees of bony erosions re observed in terms of eroded size and depth, which can be identified after plate removal.

**Figure 3 jcm-11-01026-f003:**
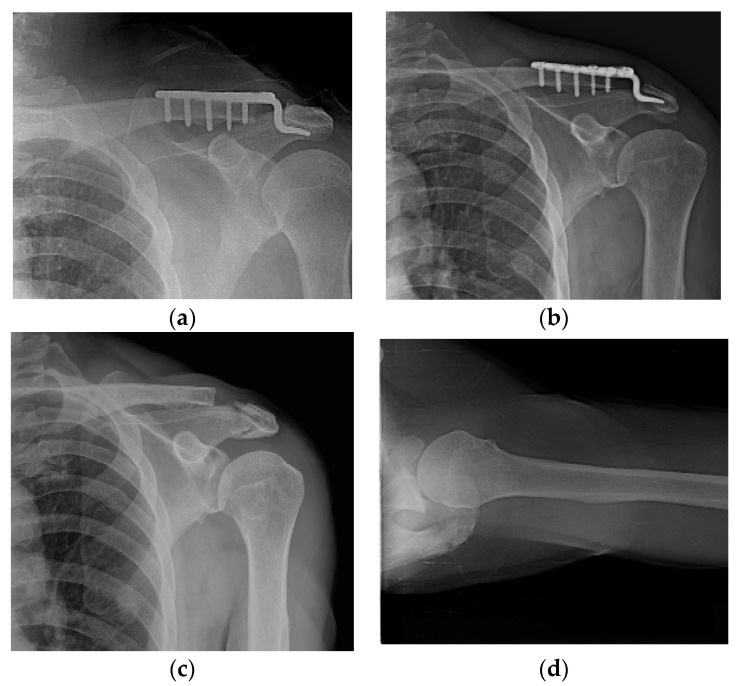
Acromial fracture. (**a**,**b**) Acromial fracture may be developed by frictional movement between the hook and the acromion; (**c**,**d**) fracture healing is shown without any additional procedure.

**Figure 4 jcm-11-01026-f004:**
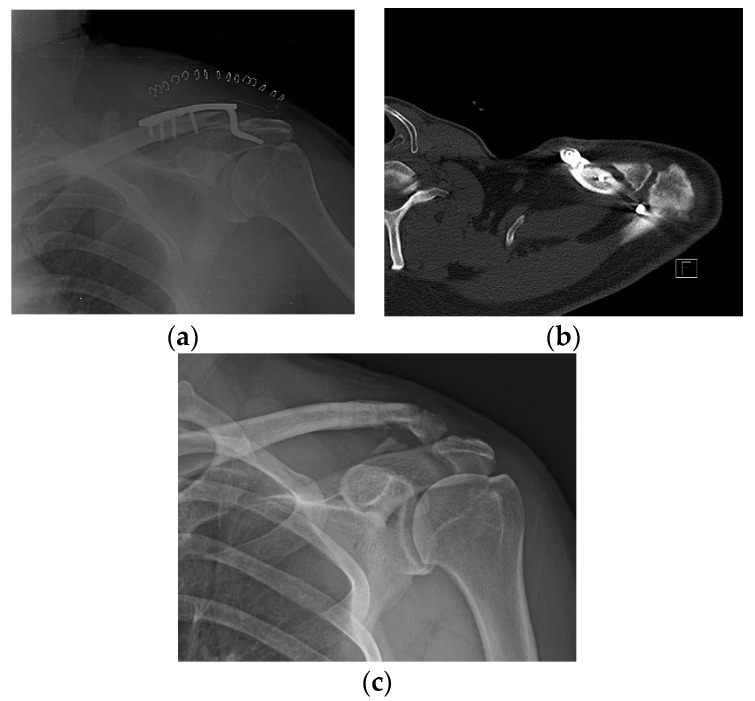
Incomplete union. (**a**) A 57-year-old male patient 4 months after hook plate fixation. (**b**) The plate is removed after incomplete union on patient request. (**c**) Bone union is achieved during follow-up.

**Figure 5 jcm-11-01026-f005:**
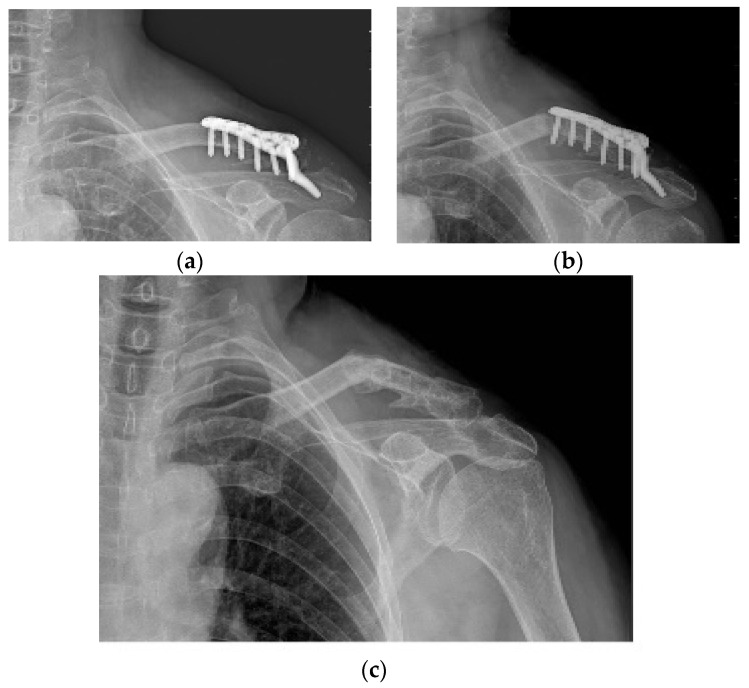
Peri-implant fracture. A 76-year-old female patient who underwent AO-type hook plating in the DCF group. (**a**) Postoperative clavicle anteroposterior radiograph view. (**b**) Patient showing peri-implant fracture on midshaft area of clavicle during the follow-up period of 4 months. (**c**) After removing the hook plate during the follow-up period, bone union is achieved.

**Figure 6 jcm-11-01026-f006:**
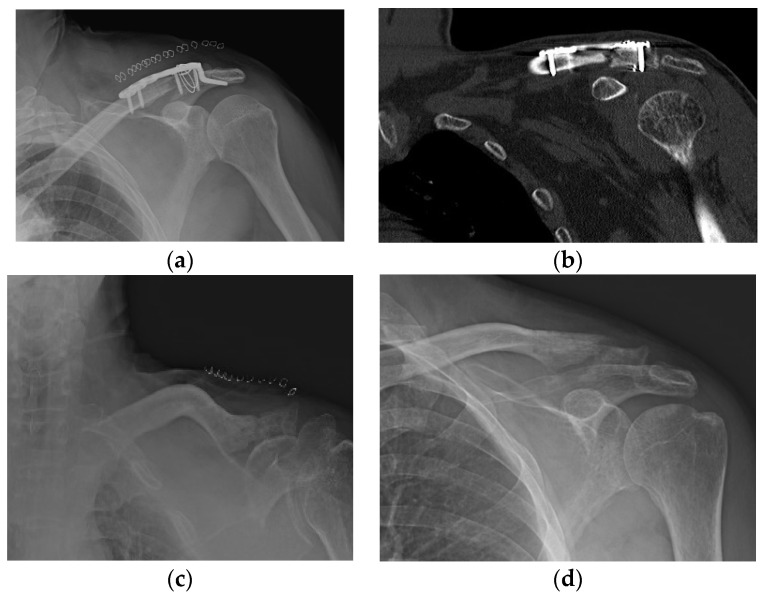
The only case of non-union among five incomplete union cases. (**a**–**c**) A 50-year-old male patient is fixed with a hook plate and bone union is not completed at 4 months, but severe painful shoulder stiffness is noted; therefore, manipulation is performed while removing the metal plate. (**d**) Bone union does not occur during the follow-up period.

**Figure 7 jcm-11-01026-f007:**
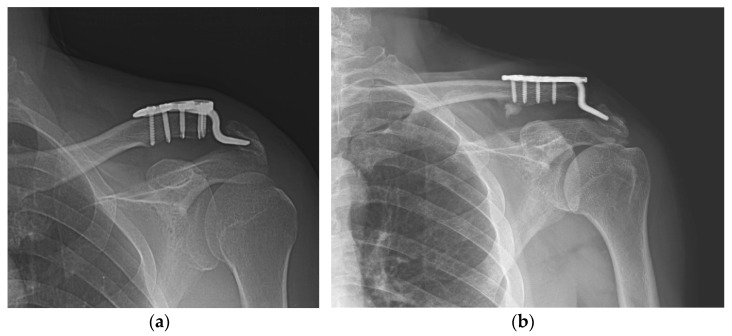
Loss of reduction in acute acromioclavicular joint dislocation. (**a**) With severe acromial erosion; (**b**) with acromial fracture.

**Table 1 jcm-11-01026-t001:** Demographics of patients receiving hook plates.

	DCF	ACD
Number of patients	76	84
Age (year)	48.47 ± 16.22	50.09 ± 13.18
Male:Female	56:20	76:8
Duration before plate removal (week)	32.7 ± 44.49	20.4 ± 8.41
Follow-up period after plate removal (month)	6.8 ± 3.3	7.4 ± 4.1

Values are presented as mean ± standard deviation. DCF: distal clavicle fracture, ACD: acromioclavicular joint dislocation.

**Table 2 jcm-11-01026-t002:** Frequency of complications using hook plate in the distal clavicle fracture and acromioclavicular joint dislocation groups.

	DCF (76)	ACD (84)	*p* Value
PSS	36 (47.4%)	33 (39.3%)	0.303
PSE	17 (22.4%)	13 (15.5%)	0.265
Acromial fracture	1 (1.3%)	7 (8.3%)	0.066
Incomplete union	5 (6.7%)		
Loss of reduction in ACD Before removal After removal		6 (7.1%) 3 (3.6%)	
Peri-implant clavicle fracture	1 (1.3%)	0	

Values are presented as number of patients. ACD, acromioclavicular joint dislocation; DCF, distal clavicle fracture; PSE, pain on shoulder elevation (>90°); PSS, painful shoulder stiffness.

**Table 3 jcm-11-01026-t003:** Frequency of painful shoulder stiffness in specific situations.

Specific Situations		Frequency of PSS	*p* Value
Fracture union status	United DCF	35 of 70 (50%)	0.362
	Incompletely united DCF	1 of 5 (20%)	
Loss of reduction	ACD with LOR	3 of 9 (33.3%)	1
	ACD without LOR	30 of 75 (40%)	
Acromial fracture	DCF with acromial fracture	1 of 1 (100%)	0.474
	DCF without acromial fracture	35 of 75 (46.7%)	
	ACD with acromial fracture	2 of 7 (28.6%)	0.699
	ACD without acromial fracture	31 of 77 (40.3%)	

DCF: the distal clavicle fracture; ACD: the acromioclavicular dislocation; LOR: the loss of reduction.

## Data Availability

Not applicable.

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
