# Peer review of "Complications and Frequency of Surgical Treatment with AO-Type Hook Plate in Shoulder Trauma: A Retrospective Study"

_jcm, 2022, doi:10.3390/jcm11041026_

Round 1

Reviewer 1 Report

This is a very intersting study regarding the use of hook plate for clavicle fractures and acromioclavicular dislocations. Althought the results are quite satisfied I would like the authors to clarify the painful shoulder stifness  (PSS) form the painful shoulder elevation (PSE). These symptoms are somewhat arbitary. So patients with painful elevation didn't have stiffness? or vise versa. 

Is it safe to inject steroid in the subacromial space, in patients with a hook plate?

How long  it takes for a clavicle fracture to heal? Usually even if union has occured radiographically we didn't remove the plate before consolidation ( 1year or more). When did these 76 clavicle fractures united?

Although you mentioned the pain intensity was measured using the VAS scale  (materials and methods , 3rd paragraph), you didn't mention the results of this scale.

Patient number 5. Figure 5c. Bone union was achieved with this angulation? 

Quality of images must be improved.

Reviewer 2 Report

Summary:

The retrospective cohort study investigates the complications resulting from treatment of AC-joint disruptions (84 cases) and distal clavicula fractures (75 cases) with an AO type hook plate. The plate was removed in all patients and follow up continued after removal. The findings are partially in accordance with the publish literature considering bony erosion of the acromion and partially differ from the publish literature considering the incidence of pain and stiffness. The authors try to distinguish between patients with painful shoulder stiffness and pain on shoulder elevation.

General concept comments:

Article:

  1. The usage of two different plates is mentioned. Can you provide some information on the difference between the two implants and the number of implantations? Does the type of implant influence the complication rate?
  2. The design of the hook and the angle between hook and acromion, as well as size and position of the hook seem to be relevant for complications. 1-5 Can you provide some information on that topic considering the used implants?
  3. Follow-up: The actual follow-up seems to be confusing. One exclusion criterium was < 3 month follow up after plate removal. The text states

“After removing the plate, postoperative follow-ups were performed at 2 weeks; 1, 2, 3, and 6 months; and 1 year.”

but Table 1 states follow up less than 1 year.

  1. The authors should try to update the literature references (only 6 references from the last 5 years). There are several studies available that compare the treatment with a hook plate with another technique, so complication rates can be found for the hook plate.6-9
  2. The refixation/reconstruction of the CC-ligaments seem to influence the complication rate. Can you provide some information on your treatment and discuss the high complication rates considering the CC ligaments? 10,11

Specific comments:

  1. “early immobilization is possible [14]” "immobilization" should be replaced by “mobilization”.
  2. “All radiographs were evaluated for osteolysis of the undersurface of the acromion by an independent examiner who was blinded to the type of fluoroscopic view taken at the time of surgery; however, subacro-mial erosion could be identified well only in the axillary radiograph of the shoulder.” This sentence seems to be from the publication [27] and therefore the information on the independent examiner could be omitted.
  3. Figure 5: The term “periprosthetic” fracture should be replaced by peri-implant fracture

References

  1. Hung L-K, Su K-C, Lu W-H, Lee C-H. Biomechanical analysis of clavicle hook plate implantation with different hook angles in the acromioclavicular joint. Int Orthop. 2017;41(8):1663-1669. doi:10.1007/s00264-016-3384-z
  2. Li G, Liu T, Shao X, et al. Fifteen-degree clavicular hook plate achieves better clinical outcomes in the treatment of acromioclavicular joint dislocation. J Int Med Res. 2018;46(11):4547-4559. doi:10.1177/0300060518786910
  3. Joo MS, Kwon HY, Kim JW. Clinical outcomes of bending versus non-bending of the plate hook in acromioclavicular joint dislocation. Clin Shoulder Elb. 2021;24(4):202-208. doi:10.5397/cise.2021.00423
  4. Xu D, Lou W, Li M, Chen J. The influence of hook tip in different depths on the acromioclavicular joint dislocation treated with clavicular hook plate: A retrospective study. Asian J Surg. 2021;44(11):1459-1460. doi:10.1016/j.asjsur.2021.07.050
  5. Qiao R, Yang J, Zhang K, Song Z. To explore the reasonable selection of clavicular hook plate to reduce the occurrence of subacromial impingement syndrome after operation. J Orthop Surg Res. 2021;16(1):180. doi:10.1186/s13018-021-02325-5
  6. Pan X, Lv R-Y, Lv M-G, Zhang D-G. TightRope vs Clavicular Hook Plate for Rockwood III-V Acromioclavicular Dislocations: A Meta-Analysis. Orthop Surg. 2020;12(4):1045-1052. doi:10.1111/os.12724
  7. Seo J, Heo K, Kim S-J, Kim J-K, Ham H-J, Yoo J. Comparison of a novel hybrid hook locking plate fixation method with the conventional AO hook plate fixation method for Neer type V distal clavicle fractures. Orthop Traumatol Surg Res. 2020;106(1):67-75. doi:10.1016/j.otsr.2019.10.014
  8. Xu Y, Guo X, Peng H, Dai H, Huang Z, Zhao J. Different internal fixation methods for unstable distal clavicle fractures in adults: a systematic review and network meta-analysis. J Orthop Surg Res. 2022;17(1):43. doi:10.1186/s13018-021-02904-6
  9. Uittenbogaard SJ, van Es LJM, Haan C den, van Deurzen DFP, van den Bekerom MPJ. Outcomes, Union Rate, and Complications After Operative and Nonoperative Treatments of Neer Type II Distal Clavicle Fractures: A Systematic Review and Meta-analysis of 2284 Patients. Am J Sports Med. 2021:3635465211053336. doi:10.1177/03635465211053336
  10. Seo J-B, Kim S-J, Ham H-J, Yoo J-S. Comparison between hook plate fixation with and without coracoclavicular ligament suture for acute acromioclavicular joint dislocations. J Orthop Surg (Hong Kong). 2020;28(1):2309499020905058. doi:10.1177/2309499020905058
  11. Chen Y-T, Wu K-T, Jhan S-W, et al. Is coracoclavicular reconstruction necessary in hook plate fixation for acute unstable acromioclavicular dislocation? BMC Musculoskelet Disord. 2021;22(1):127. doi:10.1186/s12891-021-03978-3

Round 2

Reviewer 2 Report

The manuscript has been sufficiently improved and I approve your arguments to keep it concise.